# Impedimetric Polyaniline-Based Aptasensor for Aflatoxin B_1_ Determination in Agricultural Products

**DOI:** 10.3390/foods12081698

**Published:** 2023-04-19

**Authors:** Jing Yi Ong, Sook-Wai Phang, Choo Ta Goh, Andrew Pike, Ling Ling Tan

**Affiliations:** 1Southeast Asia Disaster Prevention Research Initiative (SEADPRI), Institute for Environment and Development (LESTARI), Universiti Kebangsaan Malaysia, Bangi 43600, Malaysia; ongjy75@gmail.com (J.Y.O.); gohchoota@ukm.edu.my (C.T.G.); 2Department of Physical Science, Faculty of Applied Sciences, Tunku Abdul Rahman University of Management and Technology (TAR UMT), Jalan Genting Kelang, Setapak, Kuala Lumpur 53300, Malaysia; pinkyphang@gmail.com; 3School of Natural and Environmental Sciences, Bedson Building, Newcastle University, Newcastle upon Tyne NE1 7RU, UK; andrew.pike@ncl.ac.uk

**Keywords:** electrochemical sensor, aptasensor, polyaniline, mycotoxin, electrochemical impedance spectroscopy

## Abstract

An impedimetric aptasensor based on a polyaniline (PAni) support matrix is developed through the surface modification of a screen-printed carbon electrode (SPE) for aflatoxin B_1_ (AFB_1_) detection in foodstuffs and feedstuffs for food safety. The PAni is synthesized with the chemical oxidation method and characterized with potentiostat/galvanostat, FTIR, and UV–vis spectroscopy techniques. The stepwise fabrication procedure of the PAni-based aptasensor is characterized by means of cyclic voltammetry (CV) and electrochemical impedance spectroscopy (EIS) methods. The impedimetric aptasensor is optimized using the EIS technique, and its feasibility of detecting AFB_1_ in real sample matrices is evaluated via a recovery study in spiked foodstuffs and feedstuffs, such as pistachio nuts, cinnamons, cloves, corn, and soybeans, with a good recovery percentage, ranging from 87.9% to 94.7%. The charge transfer resistance (R_CT_) at the aptasensor interface increases linearly with the AFB_1_ concentration in the range of 3 × 10^−2^ nM to 8 × 10^−2^ nM, with a regression coefficient (R^2^) value of 0.9991 and detection limit of 0.01 nM. The proposed aptasensor is highly selective towards AFB_1_ and partially selective to AFB_2_ and ochratoxin A (OTA) due to their similar structures that differ only at the carbon–carbon double bond located at C_8_ and C_9_ and the large molecule size of OTA.

## 1. Introduction

Mycotoxins are toxic compounds produced as secondary metabolites by fungi, especially by saprophytic molds growing on foodstuffs or animal feeds. Among all the mycotoxins, aflatoxin B_1_ (AFB_1_), the carcinogenic product belonging to the flavus, parasiticus, and nomius species of the genus *Aspergillus*, is considered the most potent, naturally occurring carcinogen among all the aflatoxins [1,2]. In view of the nature of mycotoxin characteristics, which are polar and nonvolatile, high-performance liquid chromatography (HPLC) is commonly used as the standard analytical method coupled with conventional detectors, such as fluorescence (FLD), UV–vis (UV), photodiode array (PDA), and mass spectrometry (MS) (single MS and tandem MS (MS/MS)), in HPLC mycotoxin analysis [3]. In view of the limitations associated with official chromatography instrumental analytical techniques, growing demand for small devices and rapid determination, preferably under in situ conditions, has prompted the development of several probes for AFB_1_ detection, such as electrochemical immunosensors [4], enzymatic multiwalled carbon nanotubes based on aflatoxin oxidase [5], and a few aptamer-based strategies, e.g., fluorescence resonance energy transfer (FRET) biosensing using cadmium telluride (CdTe) quantum dots [1], competitive dipstick assay [6], electrochemical impedimetric assay [7], etc. However, these are characterized by poor precision and low sensitivity, stability, and selectivity.

The impedance of electrochemical impedance spectroscopy (EIS) is measured when an alternating sinusoidal potential or current is applied to the electrodes, yielding changes in the current or potential [8]. For instance, during the stimulation of a sinusoidal alternating current (AC) from a high frequency to a low frequency, the changes in potential may be caused by the process at the interface of the electrode and electrolyte, such as the charge transfer of electrons from charged atomic species to the electrode [8]. There are two categories of impedimetric biosensors based on the modification of the working electrode surface. The impedimetric biosensor is capacitive when the electrode surface is modified with a dielectric component, where the charge transfer from redox probes is inhibited. Thus, the current of the double-layer capacitance is measured with a sinusoidal voltage signal instead of a sinusoidal current signal. Since the interfacial interaction does not involve charge transfer, it is known as a non-Faradaic process [9]. During the modification of the electrode surface with a conductive component, the charge transfer of redox probes at the electrode–electrolyte interface is present. Thus, it is categorized as a Faradaic process [9]. Many impedimetric biosensors are developed based on the Faradaic process.

Generally, carbon and gold working electrodes are often used in developing an electrochemical biosensor. Although the intrinsic electrical conductivity of gold is better than carbon, a carbon working electrode is preferred due to several disadvantages of a gold working electrode. For instance, the mechanical, chemical, and electrochemical pretreatments required for gold electrodes to prepare the electrode surface for modification and detection are time-consuming. Moreover, the interfacial interaction of the gold electrode surface might be affected due to the etching caused by chloride ions (Cl^−^) in phosphate-buffered saline (PBS) and cyanide ions (CN^−^) in the ferrocyanide–ferricyanide redox probes that are commonly utilized [10]. However, gold nanomaterials are prevalently coated on the working electrode as the surface conductivity enhancer. Furthermore, it is able to increase the active surface area of the working electrode [11].

The modification of the working electrode surface is indispensable to the incorporation of the biomolecular recognition elements for the targeted substances. Therefore, the working electrode is often integrated with a layer of the component, which is capable of binding with the biomolecular recognition elements, such as antibodies, aptamers (Apts), and molecularly imprinted polymers (MIP). In most cases, the integrated component consists of the characteristic of a conductor or a semiconductor. For instance, gold nanoparticles (AuNPs) and graphene with outstanding intrinsic electrically conductive properties are often coupled in different formations during the development of a biosensor. In the work of Wen and co-workers [12], the modification of a glassy carbon electrode with AuNPs after the first layer of graphene sponge increased the active sites for homocysteine Apt attachment via covalent bonding between the gold and sulfur of the hydrogen sulfide terminal of the Apt. Furthermore, it also greatly improved the electrical conductivity of the electrode surface, thus enhancing the sensor performance during homocysteine detection. Some other examples are the nanocomposite of reduced graphene oxide (rGO) and gold nanocubes, gold-modified thiol graphene, and the nanocomposite of graphitic carbon nitride with gold nanorods [13,14,15].

The conductive polymer is an inexpensive alternative to AuNPs. Polypyrrole (PPy) is often integrated into biosensors due to various advantages, such as being electrically conductive, functionalizable, biocompatible, easily soluble in aqueous solution, and a low potential is required for electropolymerization [16]. PPy can be applied in different forms in electrochemical biosensors, such as electropolymerized film or polymer composite, during the modification of the electrode surface [16,17,18]. For instance, a nanocomposite of PPy and AuNP film is modified on the screen-printed graphite electrode (SPGE) by the electropolymerization of pyrrole to form PPy with multipulse amperometry, followed by performing cyclic voltammetry (CV) on chloroauric acid with sulphuric acid to form a layer of AuNPs on top of PPy [16]. Similarly, AuNPs have enabled the immobilization of Apt through bonding with the thiol functional group. However, it is observed that the charge transfer resistance (R_CT_) of PPy nanoparticle-coated SPGE is higher than the R_CT_ of bare SPGE. Thus, PPy does not necessarily improve the conductivity of the electrode despite being electrically conductive [16].

Another popular conducting polymer often incorporated into the electrochemical sensor is polyaniline (PAni), due to its numerous benefits, including its association with the facile synthesis procedure, high chemical and environmental stability, and most importantly, possessing high electrical conductivity [19]. Generally, PAni can be synthesized with the chemical oxidation method and electrochemical polymerization method. It can exist in different oxidation states with different forms and colors. It is known as a leucoemeraldine base when fully reduced, an emeraldine base when the leucoemeraldine base is oxidized, emeraldine salt (ES) when the emeraldine base is doped with strong acid, and a pernigraniline base when it is fully oxidized [20]. Since the electrical conductivity and capacitive properties of PAni are affected by its morphology, these properties can be adjusted by controlling the factors affecting the morphology during synthesis. Several morphologies, such as the tube, sphere, and granule shapes of PAni, can be synthesized by altering the reaction temperature during synthesis [21]. According to Kuang and co-workers [21], the shape of PAni has changed from granule to sphere and tube with the rise in synthesis temperature, and tubular PAni consists of the greatest electrical conductivity and capacitive properties among the three morphologies. The morphology of PAni can also be altered by applying different pH levels and incorporating nanomaterials, such as titanium dioxide (TiO_2_), during synthesis [22,23]. It was observed that the emeraldine salt form of PAni that is synthesized at pH 0 has a porous nanowire structure, whereas it is agglomerated in the granular form at pH 1 and pH 2, and a tubular morphology is obtained at pH 3 [24]. The porous nanowire structure, which provides a large surface area, exhibits the greatest performance in electrical conductivity, thus having the highest specific capacitance among the different PAni shapes. This is due to the higher efficiency in the diffusion of solvent molecules and ions through the structure, thus facilitating the redox process [23].

In this work, PAni is incorporated as a support matrix for the development of the impedimetric PAni-based aptasensor for AFB_1_ detection in foodstuffs and feedstuffs. In brief, PAni is synthesized by the chemical oxidation method and drop-coated onto the surface of the screen-printed carbon electrode (SPE), followed by modification with the AFB_1_ aptamer (Apt) through a crosslinking reaction with glutaraldehyde. The AFB_1_ Apt is based on a patent obtained by NeoVentures Biotechnology Inc. (Canada), which was employed in the optical studies of AFB_1_ via surface plasmon resonance (SPR) based on substrate refractive index [24], as well as fluorescence resonance energy transfer (FRET) for AFB_1_ detection based on fluorescence intensity [1]. Electrochemical interrogation of the abovementioned Apt conjugated to PAni film is first investigated in the present study. The stepwise fabrication of the PAni-based aptasensor is evaluated with the electrochemical techniques of cyclic voltammetry (CV) and electrochemical impedance spectroscopy (EIS). In addition, the electrochemical properties of the SPE surface modified with PAni film and the electroanalytical chemistry of the proposed aptasensor in determining AFB_1_ based on the charge transfer of redox probes at the electrode–electrolyte interface is discussed.

## 2. Materials and Methods

### 2.1. Apparatus and Electrodes

Chemical structure elucidation of PAni was performed with a Perkin Elmer Attenuated Total Reflectance Fourier Transform Infrared Model ATR-FTIR spectrum 100 spectrometer (PerkinElmer, Waltham, MA, USA) and Cary 50 UV–vis spectrophotomet er (Varian, Melbourne, Australia). An Autolab potentiostat/galvanostat Model M101 (Metrohm, Utrecht, The Netherlands) installed with NOVA 2.0 electrochemistry data acquisition and analysis software (version 2.0, Metrohm, Utrecht, The Netherlands) was used to perform CV and EIS measurements. Three electrode systems, comprising a rod-shaped platinum electrode, an Ag/AgCl electrode, and a screen-printed carbon electrode (SPE), were used as the counter electrode, reference electrode, and working electrode, respectively, for all of the electrochemical measurements. The Ag/AgCl electrode was filled with an internal fill solution of 3.0 M standard KCl solution. The working electrode, SPE, was purchased from Scrint Technology (M) Sdn. Bhd. (Kedah, Malaysia) and modified with PAni and Apt.

### 2.2. Chemicals

Ammonium persulfate (APS, (NH_4_)_2_S_2_O_8_) was obtained from Friendemann Schmidt (Parkwood, Australia). Dioctyl sulfosuccinate sodium salt (AOT, C_20_H_37_NaO_7_S) was purchased from Fisher Scientific (Selangor, Malaysia). Aniline (Ani, C_6_H_5_NH_2_), disodium phosphate (Na_2_HPO_4_), and glutaraldehyde (C_5_H_8_O_2_) with 25% solution in water were purchased from Acros Organic (Geel, Belgium). A 50-mer unmodified Apt of AFB_1_ and 5′-amino (standard C6) modified Apt of AFB_1_ with the base sequence of NH_2_-5′-GTT GGG CAC GTG TTG TCT CTC TGT GTC TCG TGC CCT TCG CTA GGC CCA CA-3′ were purchased from Integrated DNA Technologies (IDT, Apical Scientific Sdn. Bhd., Selangor, Malaysia). The AFB_1_ aptamer was patented by Neoventures Biotechnology Inc. (London, ON, Canada) (Patent: PCT/CA 2010/001292) [1,24]. Magnesium chloride (MgCl_2_), hydrochloric acid (HCl), toluene (C_7_H_8_), tris(hydroxymethyl)aminomethane ((Tris-HCl, NH_2_C(CH_2_OH)_3_), and zearalenone (ZEN) were procured from Sigma-Aldrich (St. Louis, MO, USA). Potassium ferricyanide (K_3_[Fe(CN)_6_]) was supplied by Nacalai Tesque (Nakagyo-ku, Kyoto). Potassium chloride (KCl) was purchased from BDH Chemicals Ltd. Sodium chloride (NaCl) and potassium dihydrogen phosphate (KH_2_PO_4_) were obtained from Merk (Selangor, Malaysia). Ethylenediaminetetraacetic acid (EDTA, C_10_H_14_N_2_Na_2_O_8_∙H_2_O) was purchased from R&M Chemicals (Semenyih, Malaysia). HPLC grade acetonitrile (CH_3_CN) was purchased from Macron Fine Chemicals^TM^ (Radnor, PA, USA). AFB_1_ (C_17_H_12_O_6_) and aflatoxin B_2_ (AFB_2_, C_17_H_14_O_6_) were obtained from ChromaDex (Los Angeles, CA, USA). Ochratoxin A (OTA, C_20_H_18_ClNO_6_) and ochratoxin B (OTB, C_20_H_19_NO_6_) were obtained from Cayman Chemical Company (Ann Arbor, MN, USA). All of the chemicals used were analytical reagent grade with a purity higher than 99%.

### 2.3. Synthesis of Polyaniline

The method of synthesizing polyaniline (PAni) adopted in the present study is similar to the method used by Karaoglan and Bindal [25], with slight modifications. PAni in solution form was synthesized via the chemical oxidation method. In this study, the PAni precipitate was extracted in the toluene instead of drying with a vacuum pump and oven after polymerization, as performed by Karaoglan and Bindal [25]. Firstly, a soft template was prepared by dissolving 6.6684 g of AOT in 200 mL of 1 M HCl solution to obtain a 0.075 M surfactant solution. AOT is a surfactant that functions to improve the solubility of PAni in both polar and nonpolar solvents, and HCl is an acidic dopant used to improve the electrical conductivity of PAni in the form of conductive emeraldine salt (ES). Next, 1.3970 g of aniline (Ani) monomer was dispersed throughout the soft template by gentle stirring, using a magnetic stirrer, at 0 °C in an ice box. After that, 0.075 M APS, which acts as an oxidant, was slowly added to the solution to initiate the oxidative polymerization reaction. The polymerization reaction proceeded for 24 h at 0 °C. Then, the resulting PAni precipitate was extracted in 150 mL of toluene to obtain PAni in solution form. The end product was rinsed three times with 100 mL of ultrapure water to remove the impurities by separating the aqueous and organic layers in a separating funnel before storage. Then, the resulting PAni precipitate was dissolved in 150 mL of toluene by mixing, using a separating funnel to obtain PAni in solution form. After the bottom aqueous layer was removed, the top layer of the PAni end product was rinsed three times with 100 mL of ultrapure water, using the same technique to remove impurities, before storage at 4 °C.

### 2.4. Preparation of Aptamer and AFB_1_ Analyte Solutions

A basic solution containing 8 mM Na_2_HPO_4_ and 136 mM NaCl was prepared by dissolving 1.136 g of Na_2_HPO_4_ and 7.96 g of NaCl in 1 L of ultrapure water. Next, 0.19 g of KH_2_PO_4_ and 0.2 g of KCl were dissolved in 1 L of ultrapure water to obtain an acidic solution with 1.4 mM KH_2_PO_4_ and 2.7 mM KCl. After that, the acidic solution was added slowly to the basic solution during continuous stirring until the phosphate-buffered saline (PBS), at 0.01 M, reached a pH of 7.5. The 50-mer 5′-amino (standard C6) modified and unmodified Apt oligonucleotides for AFB_1_ were resuspended in TE buffer at pH 7.5, which comprised of 10 mM Tris-HCl and 0.1 mM EDTA in ultrapure water. Next, the Apt suspension was incubated for 30 min at 25 °C before centrifuging at 10,000× *g* for 1 min. Then, the Apt was separated into aliquots and stored at a temperature of −20 °C. Before usage, each Apt aliquot was diluted to 5 μM in pH 7.5 folding buffer that consisted of 0.01 M phosphate buffer and 1 mM MgCl_2_. Then, aliquots were annealed at 90 °C for 5 min, followed by cooling at room temperature (25 °C) for 15 min. Subsequently, 0.016 M of AFB_1_ stock solution was prepared by dissolving 10 mg of AFB_1_ in 2 mL of acetonitrile. After that, serial dilution was performed, with PBS as the diluent, to obtain the required AFB_1_ concentration.

### 2.5. Fabrication of Aptasensor

A screen-printed carbon paste electrode (SPE) was used as the working electrode. It was combined with PAni and Apt to form a conductive biolayer on the SPE electrode surface for AFB_1_ detection. Subsequently, 4 µL of preconcentrated PAni solution was then drop-coated on the SPE working electrode and dried for 15 min at room temperature. After that, the PAni-modified SPE was immersed in 1 mL of 5% glutaraldehyde (GA) solution, which acts as a crosslinking agent in 0.01 M phosphate buffer at pH 7.5, for 1 h at ambient temperature, followed by drop-coating with 2 µL of 5 μM Apt solution, and left for 16 h, before it was sequentially rinsed with copious amounts of 0.01 M PBS (pH 7.5) and ultrapure water [7]. The Apt-modified PAni electrode was finally dried with nitrogen gas for 3 min prior to use. The fabricated impedimetric PAni-based aptasensor is illustrated in Figure 1.

### 2.6. Electrochemical Measurements

Electrochemical measurements of CV with different scan rates were performed to characterize the electrochemical properties of the PAni-based impedimetric aptasensor. The potential ranged from −1.1 V to +1.1 V versus Ag/AgCl, and the various scan rates used were 50, 100, 150, 200, 250, and 300 mV s^−1^. An open circuit potential with a single sinusoidal voltage and an input electrochemical potentiostatic impedance amplitude of 10 mV was used during the measurement at a frequency range of 1 MHz to 100 Hz. Moreover, 5 mM K_3_[Fe(CN)_6_] redox indicator in 0.01 M PBS at pH 7.5 containing 0.1 M KCl was used in all the electrochemical measurements at room temperature. During the analyses of Nyquist plots acquired from EIS measurements, all the starting points of Z′_1_ (Ω) on the real axis (*x*-axis) were standardized to 0 to facilitate better observation of the differences in the charge transfer resistance (R_CT_) of each measurement. This is due to the electrochemical response of the PAni-based impedimetric aptasensor, which is reflected only by the R_CT_; thus, the solution resistance is omitted. Therefore, only the resulting R_CT_ is provided, instead of the initial Z′_1_ (Ω), and final Z′_1_ (Ω) along with the R_CT_. During the detection of AFB_1_, 6 µL of AFB_1_ was drop-coated onto the PAni-based impedimetric aptasensor and incubated for 30 min at room temperature. After that, the aptasensor was rinsed with PBS, followed by rinsing with ultrapure water before analysis.

### 2.7. Validation of PAni-Based Impedimetric AFB_1_ Aptasensor

The feasibility of the PAni-based impedimetric aptasensor for the assay of AFB_1_ in real sample matrices was evaluated via a recovery study on spiked foodstuffs and feedstuffs. The real samples used were commercial pistachio nuts, cinnamons, cloves, corn, and soybeans. The solid real samples were ground into a powder with an electric dry mill grinder (Panasonic Blender MX-M200, Petaling Jaya, Selangor), and 2 g of sample powder was weighed into a 15 mL centrifuge tube. Next, 10 mL of the mixed solution of acetonitrile/ultrapure water at the volume ratio of 4:1 that was adjusted to pH 7.5 with 0.5 M tris-HCl, and 0.5 M tris-base was added into the centrifuge tube containing the sample powder. After sonicating (SciLab, Seoul, Korea) the heterogeneous mixture for 20 min, it was centrifuged (Thermo Fisher, Waltham, MA, USA) at 3000 rpm for 5 min to obtain the supernatant. If the samples were contaminated with AFB_1_, the AFB_1_ was extracted from the supernatant. Subsequently, 6 mL of supernatant was transferred to a 50 mL beaker and diluted with 30 mL of 0.01 M PBS. Then, 6 µL of the resultant spiked (at 0.06 nM standard AFB_1_) or unspiked solution was drop-coated onto the PAni-based impedimetric aptasensor. Distilled water was used as a negative control sample. The aptasensor response obtained in terms of R_CT_ of the real samples was substituted into the linear equation of the linear calibration curve for the AFB_1_ electrochemical aptasensor. The analytical performance of the aptasensor for AFB_1_ concentration detection was evaluated via the *t*-test, whilst recovery, precision, and accuracy of the aptasensor were calculated according to Equations (1)–(3), respectively.
(1)$$\mathrm{Recovery}\;\left(\%\right)=\frac{\mathrm{Found}\;\mathrm{Concentration}}{\mathrm{Known}\;\mathrm{Concentration}}\times 100\%$$
(2)$$\mathrm{Precision}\;\left(\%\right)=\frac{\mathrm{Standard}\;\mathrm{Deviation}\;\mathrm{of}\;\mathrm{Found}\;\mathrm{Concentration}}{\mathrm{Mean}\;\mathrm{of}\;\mathrm{Found}\;\mathrm{Concentration}}\times 100\%$$
(3)$$\mathrm{Accuracy}\;\left(\%\right)=\frac{\left|\mathrm{Found}\;\mathrm{Concentration}-\mathrm{Known}\;\mathrm{Concentration}\right|}{\mathrm{Known}\;\mathrm{Concentration}\;}\times 100\%$$

## 3. Results and Discussion

### 3.1. FTIR and UV–Vis Spectrophotometric Characterizations of PAni Substrate

Figure 2a depicts the FTIR spectrum of the PAni structure in emeraldine salt (ES) form. The analysis was performed within the wavenumber range of 650−4000 cm^−1^ using the technique of attenuated total reflection (ATR). The peak at 795 cm^−1^ is due to the C−C and C−H out-of-plane bending vibration of the benzenoid unit in the PAni backbone structure [26]. The strong peaks at 1028 cm^−1^, 1173 cm^−1^, and 1726 cm^−1^ arose from the S−O, O=S=O, and C=O groups of AOT, respectively [27,28]. This indicates the incorporation of AOT in PAni, which has proven the doping state of PAni [29]. Another distinct peak at 1206 cm^−1^ is attributed to the C−N^+•^ polaron stretching from the PAni backbone structure. In addition, the bands at 1453 cm^−1^ and 1603 cm^−1^ correspond to the C=C stretching vibration of the benzenoid and quinoid rings, respectively. Both of the strong overlapping peaks at 2856 cm^−^^1^ and 2924 cm^−^^1^ are contributed by the C−H stretching of both benzenoid and quinoid structures. The characteristic peak at 3236 cm^−1^ represents the N−H stretching benzenoid unit [30].

Figure 2b shows the UV–vis spectrum of PAni dissolved in toluene. The UV–vis absorbance was recorded in the wavelength range of 300–900 nm. Three characteristic peaks of PAni are observed at 345 nm, 425 nm, and 790 nm. At the wavelength of 345 nm, the π to π* transition of electrons from quinoid and benzenoid rings in the PAni backbone structure is observed. In addition, the peaks located at 425 nm and 790 nm are attributed to both low-energy and high-energy polaron bonds, respectively [31]. The hole polarons are created after PAni is doped by HCl, leading to the presence of polaronic excitons, which consist of both the hole polaron and electron polaron. When external energy is provided by the photons during UV–vis analysis, the electron, from its potential well of the polaron, can be excited into a higher energy state. Thus, electrons from the highest occupied molecular orbital (HOMO) will be transported to the lowest unoccupied molecular orbital (LUMO) [32]. According to Dennany and co-workers [33], the absorption at 430 nm is due to the polaronic electron band, whereas absorption at 800 nm is due to the bipolaronic electron band. Thus, the absorption peak at 425 nm is due to the excitation of electrons from the electron polaron into the antibonding π* orbital. On the other hand, the absorption at 790 nm shows the excitation of electrons from π to polaron transitions, which arose from the doping level and amount of polaron formed in PAni [34]. The broad and large band of the absorption at 790 nm shows the high conjugation structure of the PAni backbone with p orbitals and delocalized electrons, which allows the mobility of electrons, thus enabling PAni to be conductive [25]. Additionally, Figure 2b shows the successful doping of PAni since no absorption band is shown at the wavelength of 630 nm, which arose from the quinoid structure. This shows that the quinoid structure has transformed into the benzenoid structure upon doping with acid [25].

### 3.2. Electrochemical Characterization of PAni-Film-Modified Electrode

Figure 3a depicts the matrix diffusion characterization of the PAni film on the SPE. By using the K_3_[Fe(CN)_6_] redox probe in the CV study of the PAni electrode, the reduction signal of K_3_[Fe(CN)_6_] in the cathodic half-cycle, and the oxidation of the electrogenerated K_4_[Fe(CN)_6_] in the next anodic half-cycle, will appear. The changes in the redox peaks of the ferri/ferrocyanide ([Fe(CN)_6_]^3−/4−^) redox couple on the PAni-film-modified SPE with increasing scan rates from 50 mV s^−^^1^ to 300 mV s^−^^1^ during potential cycling in the range from −1.1 V to +1.1 V is shown in Figure 3a, and the detailed electrochemical data are tabulated in Appendix A. The anodic peak potential (E_pa_) has increased, whilst the cathodic peak potential (E_pc_) has decreased, with increasing scan rates. Large peak-to-peak separation (ΔE_p_) was obtained from the redox signal of Fe(CN)_6_]^3−/4−^ on the PAni-film-modified SPE by increasing the scan rate. This might have resulted from slow electron movement across the PAni film with increasing scan rate from 50–300 mV s^−^^1^. Furthermore, the cathodic peak is more obvious than the anodic peak with the PAni-modified SPE. This shows that the reduction reaction rate was higher than the rate of the oxidation reaction on the PAni-modified SPE. It is postulated that the imine functional groups of quinoids at the PAni backbone structure were being reduced by potassium ions (K^+^) at a higher reaction rate compared to the oxidation reaction. However, both the anodic peak current (i_pa_) and cathodic peak current (i_pc_) increased with the increase in scan rate. The slight anodic shift is formed from the nature of PAni [35].

Since the ΔE_p_ values at all scan rates are greater than the reversible limit of 57 mV, the redox process is quasi-reversible [36]. In addition, the E_pa_ and E_pc_ are dependent on the scan rate, which opposes the theory of a reversible system [37]. Therefore, the redox process of the charges is not completely controlled by the charge transfer step. Instead, the diffusion of charges is included [38]. Furthermore, the reversible characteristic of the PAni film can be determined based on the i_pa_ to i_pc_ ratio (i_pa_/i_pc_). Although the i_pa_/i_pc_ for scan rate at 300 mV s^−^^1^ is 1.000, the oxidation peak is not obvious. Therefore, it is more appropriate to determine the reversibility of PAni redox film with respect to ΔE_p_. Since ΔE_p_ is consistent during the reversible and rapid charge transfer process, the increasing ΔE_p_ values demonstrate a quasi-reversible characteristic of the PAni film [38]. The electroactive surface area (A_real_) of the PAni-film-modified SPE is calculated using the Randles–Ševčík equation for a quasi-reversible system, as shown in Equation (4):(4)$$i_{pa}=\pm 0.436\;nFA_{real}C\sqrt{\frac{nFDv}{RT}}$$
where i_pa_ is the anodic peak current, n is the number of electrons involved, F represents the Faraday constant, A_real_ is the electroactive surface area to be determined, C is the concentration of K_3_[Fe(CN)_6_], D is the diffusion coefficient of K_3_[Fe(CN)_6_], v is the applied scan rate, R is the molar gas constant, and T is the temperature. The v of 100 mV s^−^^1^ is used for the calculation, thus i_pa_ is 112.457 × 10^−^^5^ A. According to Gong and co-workers [19], the D for 5 mM K_3_[Fe(CN)_6_] in 0.1 M KCl is 6.30 × 10^−^^6^ cm^2^ s^−^^1^. At a T of 218.15 K, the calculated electroactive surface area of PAni-film-modified SPE is 107.97 mm^2^, which considerably increased by a magnitude of 24 compared to the bare SPE electrode at the same scan rate (4.42 mm^2^).

The matrix diffusion characteristic of the PAni film was further analyzed by using the Randles–Ševčík linear equation, i_pa_ = mv^1/2^ + c. The linear equation was obtained from a plot of current against the square root of scan rate (v^1/2^), as shown in Figure 3b. The electrochemical data of anodic and cathodic peak currents against the square root of scan rates for PAni-modified SPE is provided in Appendix A. A strong linear correlation between peak current and v^1/2^ is indicated by a high R^2^ value of 0.9897 for i_pa_ and 0.9955 for i_pc_, with the linear equation of i_pa_ (10^−^^5^ A cm^−^^2^) = 60.05v^1/2^ (V s^−^^1^)^1/2^ − 7.52 and i_pc_ (10^−^^5^ A cm^−^^2^) = −68.34v^1/2^ (V s^−^^1^)^1/2^ + 11.49, respectively. The strong relationship based on high R^2^ values proves the occurrence of electron transfer at the PAni film interface [38]. However, since the linear trendlines do not pass through the origin, the oxidation and reduction reactions have involved processes other than the diffusion of electrons [39]. According to Elugoke and co-workers [40], the occurrence points demonstrated that the charge transfer process is a diffusion-controlled and surface-confined process. The plot of anodic and cathodic peak potentials against the logarithm of scan rate for PAni-modified SPE is depicted in Figure 3c, whilst the corresponding electrochemical data are provided in Appendix A. Based on Figure 3c, the peak potential (E_p_) is dependent on log v. Since the potential of the cathodic and anodic peaks varies with log v, and the peak potential separation is large, we deduce that the redox reaction occurred with both diffusion and charge transfer kinetics [39]. Therefore, the charge transfer process in PAni-film-modified SPE involves diffusion and charge transfer kinetics, and it is a surface-confined process.

### 3.3. Electrochemical Interrogation of PAni-Film-Based AFB_1_ Aptasensor

The cyclic voltammetric study results of the stepwise fabrication of the PAni-film-based AFB_1_ aptasensor are shown in Figure 4a. The details of the respective electrochemical data are tabulated in Appendix A. All the CV scans were performed at a potential range of −1.1 V to +1.1 V. Surface modification of the SPE with PAni led to an obvious increase in SPE conductivity. Both i_pa_ and i_pc_ signals increased by 15% after the incorporation of PAni film on the SPE surface. This implies that the electroactive surface area has been enhanced by the conjugative structure provided by PAni [41]. The redox peaks’ currents of the PAni film decreased after immobilization with Apt, and further decreased after reaction with AFB_1_. The decrease in the peak current of the aptasensor indicates the success of the Apt/AFB_1_ complex formation, which restricted the charge transfer process at the sensing interface.

Surface modifications of the SPE with PAni and Apt as well as the detection of AFB_1_ were also evaluated with the electrochemical impedance spectroscopy (EIS) technique. The electrochemical EIS analysis results of the stepwise fabrication of the aptasensor and its impedimetric response with AFB_1_ in the electrolyte containing ferricyanide anions, [Fe(CN)_6_]^3−^, are shown in Figure 4b, and the corresponding R_CT_ reading is tabulated in Appendix A. The electrical characteristic of the impedance is manifested by the equivalent circuit, as shown in the inset of Figure 4b, where R_S_ is the solution resistance, R_CT_ is the charge transfer resistance, CPE represents the constant phase element, and W is the Warburg resistance arising from the transport of ions in the electrolyte to the electrode surface [42]. The charge transfer resistance can be determined from the R_CT_ of the Nyquist plot [43]. The R_CT_ at the electrode–electrolyte interface exhibits an apparent decrease after the coating of the PAni film. This shows that the conductive PAni film has enabled the rapid transport of ferricyanide ions, [Fe(CN)_6_]^3−^, at the electrode–electrolyte interface, thus decreasing the R_CT_ [41]. The conductivity of PAni is mainly contributed by the C−N^+•^ polaron lattice (PL) structure, which is formed from the doping of the imine functional group in quinoids [44]. The charge transfer is directly affected by the conductive parts of PAni [45,46]. Therefore, the sensitivity of the electrode was enhanced. Subsequently, the R_CT_ increased after the immobilization of Apt. This was due to the electrostatic repulsion between the negatively charged phosphate group of Apt and the anionic [Fe(CN)_6_]^3−^ ion; thus, the charge transfer process was hindered. Furthermore, Apt is nonconductive in nature [42,47]. This indicates the successful incorporation of Apt via a glutaraldehyde crosslinking agent. Schiff bases were formed at the terminal amine group of PAni, where the polar covalent bond was formed between the carbon atom of the aldehyde group and the nitrogen atom of the amide group [48,49]. The R_CT_ was further enhanced after AFB_1_ detection. This resulted from the resistance in charge transfer of the PAni film after AFB_1_ had formed a complex with the immobilized Apt via shape complementarity at the binding interface [37]. Apt formed a hairpin structure with a loop after being annealed at 90 °C, and formed an active site for AFB_1_ [7]. In addition, the formation of the Apt/AFB_1_ complex might lead to a steric hindrance effect, which could hamper the process of charge transfer at the electrode–electrolyte interface [41].

### 3.4. Optimization of Electrochemical PAni-Modified Aptasensor for Impedance Detection of AFB_1_

#### 3.4.1. Optimizing the Volume of PAni on the SPE Electrode

The effect of PAni volume deposited on the SPE during the fabrication of the aptasensor is shown in Figure 5a. Based on the electrochemical data tabulated in the Appendix A, the R_CT_ for SPE modified with 4 µL PAni is smaller than that for SPE modified with 3 µL PAni. This implies that the transportation rate of the negatively charged [Fe(CN_6_)]^3−^ ions to the electrode interface was increased through using a larger volume of electrically conductive PAni matrices [41,50]. The resultant sensitivity of the electrode was enhanced with a smaller R_CT_ value. Thus, 4 µL of PAni was used to fabricate the aptasensor for AFB_1_ detection in the following aptasensor optimization studies.

#### 3.4.2. Sensitivity of the PAni-Film-Based Impedimetric Aptasensor towards AFB_1_ Detection

The sensitivity of the PAni-based aptasensor towards AFB_1_ was determined by examining the impedimetric responses of the aptasensor upon reaction with different AFB_1_ concentrations, ranging from 0.03 nM to 0.25 nM. The Nyquist plots of the detection of AFB_1_ in the concentration range of 0.03–0.25 nM using the aptasensor are illustrated in Figure 5b. The R_CT_ of triplicate impedimetric detection of AFB_1_ is tabulated in Appendix A. According to Figure 5c, the PAni-based aptasensor reached a saturation state during the detection of 0.08 nM AFB_1_. The decrease in the R_CT_ response of the aptasensor during the detection of 0.10 nM and 0.12 nM AFB_1_ is attributed to the equilibrium displacement phenomenon, whereby the high loading of AFB_1_ on the aptasensor causes the detachment of immobilized AFB_1_ Apts from the PAni-modified surface [51]. This resulted in a significant reduction in the R_CT_ response of the aptasensor as the electrically conductive PAni film was exposed to the electrolyte to allow a higher transport rate of [Fe(CN_6_)]^3−^ ions at the electrode–electrolyte interface. The detachment rate of immobilized AFB_1_ Apt reached a constant state upon reaction with 0.12 nM AFB_1_ and beyond. The R_CT_ responses of the aptasensor towards 0.12–0.25 nM AFB_1_ fluctuate within a narrow range from 205–207 Ω, which is lower than the intrinsic R_CT_ for the PAni/Apt-modified aptasensor, and is similar to that of the PAni-modified SPE’s R_CT_ response. This indicates that only a small amount of the immobilized Apt was detached from the aptasensor’s surface.

The linear calibration range of the PAni-based aptasensor was determined to be in the range of 0.03 nM to 0.08 nM, with a linear equation of y = 743.98x + 190.52 and an R^2^ value of 0.9991, which indicates a significant linear correlation between the R_CT_ and the concentration of AFB_1_, as shown in the inset of Figure 5c. The reproducibility relative standard deviations (RSD) of the aptasensor were generally lower than 5.0%, which indicates great reproducibility of the aptasensor in AFB_1_ detection. The limit of detection (LOD) was estimated to be 0.01 nM with an RSD of 1.3%, determined using Equation (5).
(5)$$LOD=\frac{3.3\times Standard\;Deviation}{Gradient}$$

The content of AFB_1_ in foodstuffs and feedstuffs at the highest acceptable level is regulated by the maximum levels (MLs), which indicate their safeness for consumption. The ML of AFB_1_ in foodstuffs such as grains, nuts, dried fruit, and spices is 2.0–8.0 μg kg^−1^ (6.40–25.62 nM), 2.0–12.0 μg kg^−1^ (6.49–38.43 nM), and 5.0 μg kg^−1^ (16.01 nM) according to the Commission Regulation (EC) No 1881/2006 and 1126/2007, Sanitation Standard for Contaminants and Toxins in Food of the Republic of China, and Food Regulations of Singapore, respectively [52,53,54,55]. On the other hand, cereal-based products intended for infants and children and dietary foods for special medical purposes have a lower ML of 0.1 μg kg^−1^ (0.32 nM), as provided by the Commission Regulation (EC) No 1881/2006 and 1126/2007, Sanitation Standard for Contaminants and Toxins in Food of the Republic of China, Food Regulations of Singapore, and Food Regulations 1985 of Malaysia, respectively. Furthermore, the ML of AFB_1_ in feedstuffs such as cereals, cereal products, maize by-products, and complementary and complete feedstuffs provided in the Commission Recommendation 2006/576/EC and Directive 2002/32/EC is in the range of 5.0–20.0 μg kg^−1^ (16.01–64.05 nM) [56,57]. Although the detection range of the PAni-based aptasensor (0.03–0.08 nM) is lower than the MLs of AFB_1_ for foodstuffs and feedstuffs (0.32–64.05 nM), it is a highly sensitive screening technique for the monitoring of AFB_1_ content in foodstuffs and feedstuffs. Therefore, it is ideal for the early and sensitive detection of AFB_1_ contamination, and appropriate precautions can be taken to avoid reaching the MLs of AFB_1_ in foodstuffs and feedstuffs. For real samples with AFB_1_ concentrations above the upper limit of quantitation of the proposed PAni-based impedimetric aptasensor, appropriate dilution of the sample using 0.01 M PBS as the diluent is needed before detection to minimize the background noise.

#### 3.4.3. Response Time of PAni-Film-Based AFB1 Aptasensor

The response time of the PAni-based aptasensor is optimized by varying the conjugation time of immobilized Apt with 0.05 nM AFB_1_. Based on the impedimetric response of the aptasensor in Appendix A, the optimized conjugation time for the aptasensor to bind with target AFB_1_ via exquisite shape complementarity was 30 min, with an RSD lower than 2.0%. This is because the R_CT_ response during the formation of Apt/AFB_1_ complexes increased at the conjugation time of 30 min. Therefore, the Apt/AFB_1_ conjugation time of 30 min was used during the electrochemical detection of AFB_1_. It was observed that 15 min of conjugation time was insufficient to obtain the maximum degree of interaction between immobilized Apt on the electrode surface and AFB_1_. On the contrary, a long incubation time of 45 min and 60 min may be unfavorable for the formation of Apt/AFB_1_ complexes. This occurrence was also present in other aptasensor evaluation research [58,59]. The resultant R_CT_ values of the aptasensor at 45 min and 60 min are obtained under conditions where minimum Apt/AFB_1_ complexes were formed, since their responses are similar to the R_CT_ of the PAni/Apt-modified SPE before AFB_1_ detection. This may be due to the instability of the Apt structure after the formation of Apt/AFB_1_ complexes. In the molecular dynamic simulations study of Mousivand and co-workers [60], the structure of Apt underwent conformational changes, with the formation of a gap between subsequent base pairs of the Apt due to the intercalation of AFB_1_. The Apt sequences applied in their simulation study have the exact same sequences as the Apt used in the present research. Therefore, it is postulated that the formation of Apt/AFB_1_ complexes disintegrated after 30 min of incubation. Since the stability of the Apt structure decreased after the intercalation of AFB_1_, the binding affinity of Apt towards AFB_1_ was weakened. Therefore, the minimum amount of AFB_1_ was detected at 45 min and 60 min.

#### 3.4.4. Selectivity of the PAni-Film-Based Electrochemical Aptasensor

The impedimetric responses of the PAni-based aptasensor during the detection of 0.05 nM AFB_1_, AFB_2_, OTA, OTB, and ZEN are illustrated in Figure 5d. The electrochemical aptasensor is highly selective towards AFB_1_, whilst being partially selective to AFB_2_ and OTA, since the molecular structures of AFB_1_ and AFB_2_ are highly similar, with an absence of the carbon–carbon double bond at C_8_ and C_9_ in the terminal furan ring of AFB_2_. It is postulated that some Apt/AFB_2_ complexes were formed due to similar molecular structures, since Apt binds to the target with exquisite shape complementarity at the PAni/Apt-modified electrode surface. Moreover, the R_CT_ response from the detection of OTA was deduced to be attributed to the electrostatic repulsion and steric hindrance between both the negatively charged Apt and OTA that restrained the transport of [Fe(CN)_6_]^3−^ ions to the electrode surface [61,62]. It is postulated that the greater molecular size of OTA (403.81 g mol^−1^) compared to AFB_1_ led to the situation where some of the OTA molecules became trapped within the PAni-modified electrode surface, since the aptasensor surface was not passivated in the way that the gold electrode surface was passivated with 6-mercaptohexanol [63]. Therefore, some of the OTA molecules remained trapped, although the aptasensor was rinsed with PBS and ultrapure water before the impedimetric measurement.

#### 3.4.5. Long-Term Stability of the PAni-Film-Based Aptasensor

The long-term stability of the aptasensor was determined by detecting 0.05 nM AFB_1_ with the fabricated PAni-based aptasensors that were stored at 4 °C over 20 days in dry conditions without immersing in PBS buffer solution. The resulting aptasensor stability represented with R_CT_ is illustrated in Appendix A. The PAni-based aptasensor demonstrated good stability within 5 days of storage at 4 °C. The average aptasensor’s R_CT_ signal of 228.176 Ω during day 1 of AFB_1_ detection decreased by 3.7% on day 5 and 10.5% on day 20. The Apt immobilized on the aptasensor gradually detached from the aptasensor after day 6 of storage. The average aptasensor signal of 209.924 Ω on day 6 was lower than the signal of the freshly fabricated aptasensor (213.756 Ω) and higher than the impedimetric signal of the PAni-modified SPE (186.338 Ω). The decrement in R_CT_ reflects the detachment of Apt from the aptasensor. Therefore, lower levels of Apt were present on the aptasensor surface to form Apt/AFB_1_ complexes during AFB_1_ detection, resulting in a low aptasensor signal. The detachment of Apt might be due to the disintegration of the Schiff bases’ imine bonds (C=N), which act as a crosslinking agent for PAni and Apt. This is postulated due to the possibility of the reversible nature of the dehydration reaction to form Schiff bases after extended exposure to buffer solution at high pH levels [64]. The stability of the Schiff bases was weakened from time to time; thus, more Apts were detached from the aptasensor surface, resulting in a further decrease in the aptasensor signal on day 9 and day 20 during AFB_1_ detection.

#### 3.4.6. Repeatability of the PAni-Film-Based Impedimetric Aptasensor

The repeatability study was performed by first detecting 6 µL of 0.05 nM AFB_1_ with the impedimetric PAni-film-based aptasensor for 30 min, and the PAni-based aptasensor was rinsed with PBS followed by ultrapure water. After the first impedimetric measurement of the aptasensor was taken, it was regenerated with 10 µL of 50 µM unmodified Apt of AFB_1_ for 30 min, and the second impedimetric quantification of 0.05 nM AFB_1_ was performed after rinsing with PBS and ultrapure water. The aptasensor was regenerated by applying the concept of equilibrium displacement, using a more concentrated unmodified Apt to create a competitive binding interaction with the immobilized Apts towards their target AFB_1_ molecules [51]. As the dispersed unmodified Apt in solution is not attached to the surface of PAni-film-based aptasensor, it will be rinsed away as a free Apt/AFB_1_ complex during the aptasensor regeneration procedure.

Based on the electrochemical data that are tabulated in Appendix A, the PAni-based aptasensor exhibited high reversibility performance over four successive detection cycles, with a repeatability RSD lower than 5% (*n* = 4). Therefore, the PAni-based aptasensor is regenerable, and is ready to bind with AFB_1_ molecules again. According to Figure 5e, although the R_CT_ of each aptasensor increased after each regeneration cycle, no significant deviation of the resultant R_CT_ was observed, since the repeatability RSD was lower than 5% (*n =* 4). This shows that the sensing platform of the aptasensor is highly stable. Since the regeneration study involves repetitive EIS measurement on the same individual PAni-based aptasensor, there is a possibility that the PAni film will experience electrochemically induced aging. The stability of the PAni film might be contributed by the immobilized layer of Apt on top of the PAni film. This is due to the properties of Apts, which are full of anionic charges arising from the phosphate group, thus forming an anionic shield on the PAni/Apt aptasensor surface [65]. Therefore, the anionic shield on the PAni/Apt aptasensor surface electrostatically repelled the [Fe(CN)_6_]^3−^ ion in the electrolyte, thus reducing the oxidizing effect from the [Fe(CN)_6_]^3−^ ions, which would lead to the degradation of the PAni film.

### 3.5. Comparison of the Analytical Performance of the Developed PAni-Film-Based Impedimetric Aptasensor with Other Electrochemical Sensors/Biosensors for AFB_1_ Detection

The performance of the proposed PAni-based aptasensor in AFB_1_ detection is here compared to other existing electrochemical sensors, as summarized in Table 1. The LOD of the developed PAni-based aptasensor is lower than the electrochemical sensor of Krittayavathananon and Sawangphruk [66] and Chokkareddy and Redhi [67], although the linear range of the PAni-based aptasensor is limited. Despite the fact that the electrochemical aptasensor produced by Jahangiri–Dehaghani and co-workers [68] exhibits excellent performance with indirect detection of AFB_1_, the fabrication method is complicated and time-consuming for a disposable aptasensor. In brief, polishing the glassy carbon electrode is necessary before modification. After the modification of AuNP/nickel-based metal–organic framework nanosheets on the carbon electrode by drop-coating, 10 cycles of the potential scan are required for 3-mercaptopropionic acid electrodeposition before the modification with Apt and bovine serum albumin (BSA). After that, cDNA is hybridized with the immobilized Apt for 1 h, followed by another 1 h of incubation with 4,4′-biphenol (PBP) before 2 h of incubation with AFB_1_ for detection [68]. Similarly, the fabrication method of the MIP-based electrochemical sensor of Singh and co-workers [69] is difficult and requires long hours of preparation, despite the extremely low LOD and wide detection range of AFB_1_. For example, hydrolysis of the indium tin oxide (ITO)-coated glass substrate with ammonia and hydrogen peroxide solution for 1 h at 80 °C is required before electrophoretic deposition of the PAni-based MIP. Additionally, ultrasonication of MIP in isopropyl alcohol for 10 h is required prior to the electrophoretic deposition. Furthermore, the raw material used, such as indium tin oxide, is expensive. Similarly, the AuNPs in the electrochemical immunosensor presented by Liu and co-workers [70] are expensive as well. Unlike the existing electrochemical sensor, the PAni-based aptasensor is inexpensive with a low LOD of 0.01 nM, high repeatability of up to four times, good stability within 5 days of storage at 4 °C, and a facile fabrication method.

To improve the sensitivity over a wide range of analyte mycotoxin, recent studies on aptasensors with advanced and emerging techniques have been reported based on an in-plane AnNPs–black phosphorus heterostructure (AuNPs-BPNS) was synthesized for signal amplification with a dual-signaling strategy, in which a methylene-blue-labeled aptamer and ferrocene monocarboxylic acid in the electrolyte was fabricated for sensitive detection of patulin [71]. Yao et al. [72] developed a dual-enzyme-based signal-amplified aptasensor with emerging CRISPR-based technology to sensitively detect zearalenone by using two aptamers as recognition elements and EnGen LbaCas12a and Nt.AlwI nicking endonuclease as signal amplifiers. Guo’s research group [73], on the other hand, made use of strand displacement amplification primers as an amplification approach initiated under the action of *Bst* DNA polymerase and nicking endonuclease for real-time fluorescence aptasensing of ochratoxin A, with amplified ssDNA products dyed with SYBR Green II. A highly sensitive and specific aptasensor has been developed by Wang et al. [74] for the anti-interference detection of OTA. In brief, 4-[(trimethylsilyl) ethynyl] aniline (4-TEAE) and OTA aptamer were assembled on the AuNPs to serve as anti-interference SERS probes, with Fe_3_O_4_ nanoparticles linked to the complementary aptamer. The specific binding of OTA to the aptamer inhibited the binding of the SERS probes and capture probes and attenuated the Raman responses.

### 3.6. Detection of AFB_1_ in Real Samples with the PAni-Film-Based Aptasensor

The performance of the PAni-based aptasensor during AFB_1_ detection in real samples, including corn, soybean, pistachio nut, cinnamon, and clove, are tabulated in Table 2. The PAni-based aptasensor exhibited a good recovery percentage ranging from 87.9% to 94.7%. This is comparable to the other electrochemical sensors applied in AFB_1_ detection, as tabulated in Table 1, such as the EIS sensor from the work of Krittayavathananon and Sawangphruk [66], which has a recovery percentage of 86.0% to 96.0%. A two-tailed *t*-test was used to evaluate the differences between the mean values of found AFB_1_ concentration in the real samples that were spiked with 0.06 nM AFB_1_ and the standard AFB_1_ solution with a concentration of 0.06 nM. The resulting AFB_1_ detection exhibited very low dispersion, with *t* values of 0.001 and 0.002 at a confidence level of 95% and *t*_critical_ value of 2.776. Since the *t* values of 0.001 and 0.002 fall between *t*_critical_ values of −2.776 and +2.776, the null hypothesis of the *t*-test was accepted, and no differences were found between the two groups of mean values. The accuracy of the result was determined through the percent relative error (bias%) between the measured and spiked concentration of AFB_1_ [75]. Based on Table 2, the accuracy and precision of PAni-based aptasensor are acceptable, since they are lower than 20.0% for impedimetric detection of low AFB_1_ concentration [76]. Thus, the PAni-based impedimetric aptasensor is validated to be applicable in AFB_1_ detection.

## 4. Conclusions

In summary, a facile impedimetric PAni-based aptasensor was developed for AFB_1_ detection in foodstuffs and feedstuffs, such as pistachio nut, cinnamon, clove, soybean, and corn, with a recovery percentage of 87.9% to 94.7% within the linear detection range of 0.03 to 0.08 nM, and with an R^2^ value of 0.9991 and LOD of 0.01 nM. The aptasensor has high selectivity for AFB_1_, and it is partially selective to AFB_2_ and OTA due to the similar structure of AFB_1_ and AFB_2_ and the large molecular structure of OTA. The coating of the PAni film on the SPE surface has improved the sensitivity of the aptasensor by facilitating the charge transfer of [Fe(CN_6_)]^3−^ ion during impedimetric measurement. The aptasensor has high reusability of four consecutive impedimetric AFB_1_ detections, with a repeatability RSD lower than 5% (*n =* 4). In addition, it has a response time of 30 min and exhibits good stability for 5 days of storage at 4 °C. Therefore, the developed PAni-based aptasensor is suitable as a cost-effective disposable sensor for monitoring the presence of AFB_1_ in foodstuffs and feedstuffs, since the raw materials of the fabricated aptasensor are inexpensive and its fabrication method is simple.

## Figures and Tables

**Figure 1 foods-12-01698-f001:**
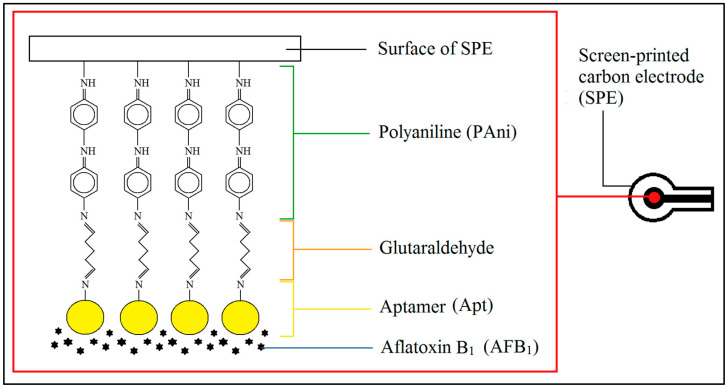
Schematic illustration of the impedimetric PAni-based aptasensor for AFB_1_ detection.

**Figure 2 foods-12-01698-f002:**
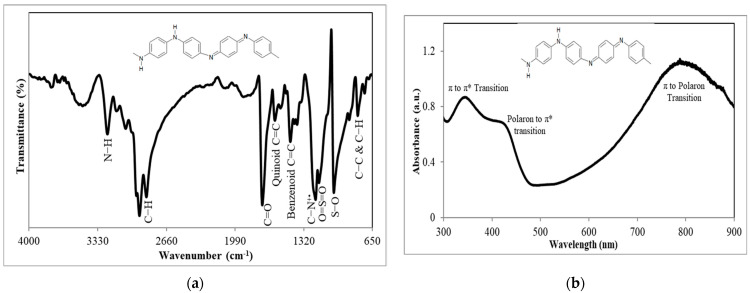
(**a**) FTIR spectrum of chemical oxidative polymerized PAni measured by Attenuated Total Reflectance Fourier Transform Infrared (Perkin Elmer model FTIR spectrum 100 spectrometer). (**b**) UV–vis spectrum of chemical oxidative polymerized PAni captured by UV–vis spectrophotometer (Varian Cary 50). The chemical structure of PAni in ES form is shown in the inset.

**Figure 3 foods-12-01698-f003:**
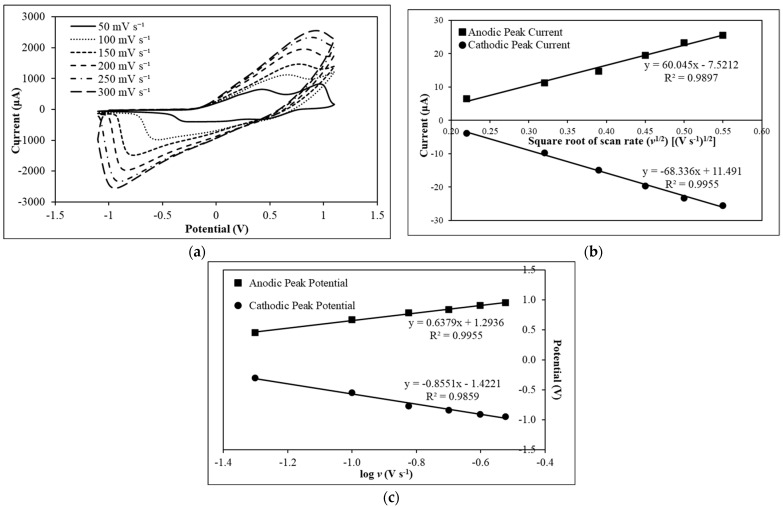
(**a**) Cyclic voltammograms of PAni-modified SPE in 5 mM K_3_[Fe(CN)_6_] redox indicator containing 0.1 M KCl at different scan rates of 50 mV s^−^^1^, 100 mV s^−^^1^, 150 mV s^−^^1^, 200 mV s^−^^1^, 250 mV s^−^^1^, and 300 mV s^−^^1^. (**b**) Plot of anodic and cathodic peak currents against the square root of scan rate for PAni-modified SPE. (**c**) Plot of anodic and cathodic peak potentials against the logarithm of scan rate for PAni-modified SPE.

**Figure 4 foods-12-01698-f004:**
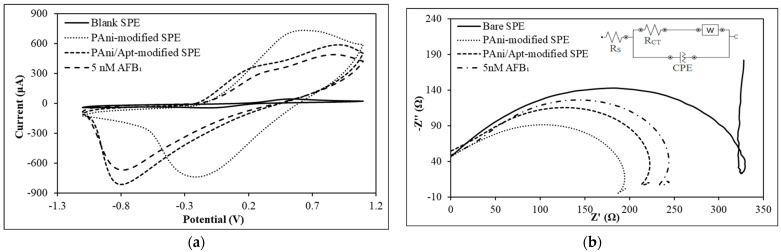
(**a**) Cyclic voltammograms of the bare SPE, PAni-modified SPE, PAni/Apt-modified SPE, and PAni/Apt-modified SPE during 5 nM AFB_1_ detection at a scan rate of 100 mV s^−^^1^ in the presence of 5 mM K_3_[Fe(CN)_6_] redox species. (**b**) Nyquist plots of bare SPE, PAni-modified SPE, PAni/Apt-modified SPE, and PAni/Apt-modified SPE during 5 nM AFB_1_ detection in 13 mL of 5 mM K_3_[Fe(CN)_6_] redox indicator containing 0.1 M KCl at 25 °C.

**Figure 5 foods-12-01698-f005:**
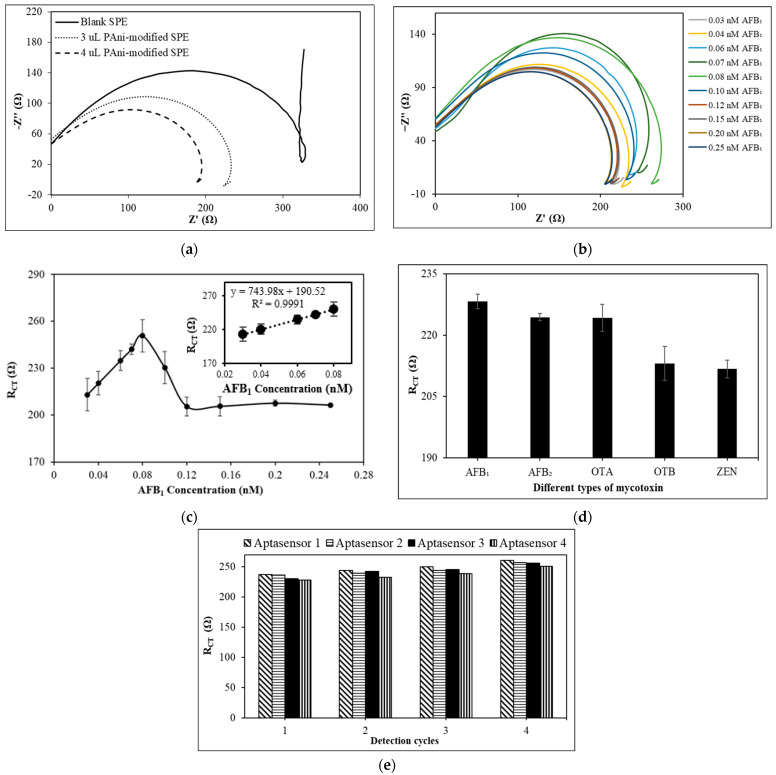
(**a**) Nyquist plots for bare SPE and SPE modified with 3 μL and 4 μL of PAni in 5 mM K_3_[Fe(CN)_6_] redox indicator containing 0.1 M KCl. (**b**) Nyquist plots of PAni/Apt SPE during the detection of varying AFB_1_ concentration ranging from 0.03 nM to 0.25 nM in 5 mM K_3_[Fe(CN)_6_] redox indicator containing 0.1 M KCl at pH 7.5. (**c**) The dynamic range of the aptasensor. The inset is the linear calibration range between 0.03 nM and 0.25 nM AFB_1_ in 5 mM K_3_[Fe(CN)_6_] redox indicator containing 0.1 M KCl at pH 7.5. (**d**) Bar graph of the PAni-based aptasensor during the detection of AFB_1_ with various interference species, such as AFB_2_, OTA, OTB, and ZEN, in 5 mM K_3_[Fe(CN)_6_] redox indicator containing 0.1 M KCl at pH 7.5. (**e**) Bar chart of the reversibility of the AFB_1_ aptasensor during 0.05 nM AFB_1_ detection (30 min); regeneration of the aptasensor is achieved by incubating the aptasensor in 50 μM unmodified AFB_1_ Apt solution as the regeneration solution for 30 min. The error bars indicate ± standard deviation, *n* = 3.

**Table 1 foods-12-01698-t001:** The comparison of different electrochemical sensors for the determination of AFB_1_ in food samples.

EC Technique	Types of Working Electrode	Electrode Surface Modification	Mycotoxin	Real Sample	LOD (nM)	Linear Range (nM)	Recovery (%)	Reference
EIS	Glassy carbon rotating disk electrode	Reduced graphene oxide aerogel (rGO_ae_) Single stranded thiol-modified DNA (ss-HSDNA)	AFB_1_	UHT milk	0.128	0.320–224.158	86.0–96.0	[66]
DPV	Glassy carbon electrode	Methyltrioctylammonium chloride ionic liquid (IL) Fe_3_O_4_ nanorods rGO	AFB_1_	Ground paprika	0.096	0.064– 1.057	98.6–101.6	[67]
DPV	Glassy carbon electrode	AuNP Nickel-based metal–organic framework nanosheets (Ni-MOF) 3-Mercaptopropionic acid (MPA) Apt cDNA 4,4′-Biphenol (PBP)	AFB_1_	Rice flour	0.003	0.016–480.338	98.4–101.3	[68]
DPV	Glassy carbon electrode	Porous AuNP Specific peptide	AFB_1_	Glutinous rice, corn, rice	0.003	0.032– 64.045	88.4–102.0	[70]
DPV	Indium tin oxide (ITO)-coated glass	PAni film as MIP	AFB_1_ FUM B_1_	Corn	0.001	0.003–1601.128 0.001– 692.684	91.6–108.6 86.2–102.4	[69]
EIS	Screen-printed carbon paste electrode (SPE)	PAni film Glutaraldehyde Apt	AFB_1_	Pistachio nut, cinnamon, clove, soybean, corn	0.010	0.030– 0.080	87.9–94.7	This work

**Table 2 foods-12-01698-t002:** The validation of the PAni-based aptasensor in AFB_1_ detection in various types of real samples such as pistachio nut, cinnamon, and clove as foodstuffs and corn and soybeans as feedstuffs.

Type of Samples	Standard AFB_1_ Concentration Spiked into Samples (nM)	Mean of Found AFB_1_ Concentration (nM), *n* = 3	Mean of Aptasensor R_CT_ Response (Ω), *n* = 3	*t* Value (At CL = 95%, *t*_critical_ = 2.776)	Recovery (%)	Precision (%)	Accuracy (%)
Negative sample (distilled water)	-	0.030 ± 0.005	212.774 ± 3.898	-	-	-	-
Corn	-	0.038 ± 0.003	218.875 ± 1.931	-	-	7.9	-
Pistachio nut	0.06	0.053 ± 0.003	229.755 ± 2.278	0.002	87.9	5.8	12.1
Cinnamon	0.06	0.057 ± 0.008	232.804 ± 5.696	0.001	94.7	13.5	5.3
Clove	0.06	0.054 ± 0.004	230.365 ± 3.011	0.002	89.3	7.6	10.7
Soybean	0.06	0.054 ± 0.006	230.394 ± 4.102	0.001	89.3	10.3	10.7
Corn	0.06	0.054 ± 0.005	230.588 ± 3.638	0.001	89.8	9.1	10.2

## Data Availability

The data presented in this study are available on request from the corresponding author.

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
