# Peer review of "Impedimetric Polyaniline-Based Aptasensor for Aflatoxin B_1_ Determination in Agricultural Products"

_foods, 2023, doi:10.3390/foods12081698_

Round 1

Reviewer 1 Report

The paper presents an impedance aptasensor for Aflatoxin B1 detection. The paper is generally well written with discussions on the achieved results. The aptasensor has been tested with real food samples and tests of repeatability and specificity have been carried out. I think it needs minor revisions:

-       1) The main problem of the proposed aptasensor is the limited working range, as also presented in Table 1. As discussed at lines 578-597, real food samples can be characterized by AFB1 concentrations much higher than values detectable with the proposed aptasensor. The authors should explain how real samples concentrations of AFB1 can be measured with the proposed aptasensor. Maybe multiple dilutions of the sample should be tested?

-          2) Line 188: “reached a pH of pH 7.5” should be changed to “reached a pH of 7.5”

-          3) The title of subsection 3.1 is missing.

-         4) In the caption of Fig. 5 and in the text (line 571) it is mentioned an inset of Figure 5c. However, no inset is present in Figure 5c.

-      5) In the first two lines of Table 2 the spiked concentration (column 2) is reported to be 0.00 nM. Is this correct?

The English language is generally good with minor typos to correct.
I think it can be accepted after minor revisions.

Author Response

The paper presents an impedance aptasensor for Aflatoxin B1 detection. The paper is generally well written with discussions on the achieved results. The aptasensor has been tested with real food samples and tests of repeatability and specificity have been carried out. I think it needs minor revisions:

1) The main problem of the proposed aptasensor is the limited working range, as also presented in Table 1. As discussed at lines 578-597, real food samples can be characterized by AFB1 concentrations much higher than values detectable with the proposed aptasensor. The authors should explain how real samples concentrations of AFB1 can be measured with the proposed aptasensor. Maybe multiple dilutions of the sample should be tested?

ANSWER- For real samples with AFB1 concentration above the upper limit of quantitation of the proposed PAni-based impedimetric aptasensor, appropriate dilution of the sample using 0.01 M PBS as the diluent is needed before detection to minimize the background noise. The correction has been done from lines 607-610 of the revised manuscript.

2) Line 188: “reached a pH of pH 7.5” should be changed to “reached a pH of 7.5”

ANSWER- Correction has been done accordingly. Please refer to line 207 of the revised manuscript.

3) The title of subsection 3.1 is missing.

ANSWER- The title of subsection 3.1, i.e. ’FTIR and UV-vis spectrophotometric characterizations of PAni substrate’ has been included in the revised manuscript accordingly.

4) In the caption of Fig. 5 and in the text (line 571) it is mentioned an inset of Figure 5c. However, no inset is present in Figure 5c.

ANSWER- The inset of Figure 5c has been included in the revised manuscript accordingly.

5) In the first two lines of Table 2 the spiked concentration (column 2) is reported to be 0.00 nM. Is this correct?

ANSWER- Because distilled water (negative control) and one of the corn samples were unspiked samples and there should be no figures of standard AFB1 concentration recorded in the first two lines of Table 2, therefore 0.00 nM has been replaced with the symbol dash ‘-’ in Table 2.

Reviewer 2 Report

An Impedimetric PAni-based aptasensor was developed to detect aflatoxin B1 in food and feed in various commodities. The selectivity for AFB1 was established. The method had a responsible limit of detection and limit of quantitation. In addition, its cross-reactivities with OTA and AFB2 were determined.  The materials developed have been fully characterized.

The introduction covers relevant material but could be expanded. The materials and methods are clearly described and reproducible. The discussion section is comprehensive but could be expanded to discuss work pertinent to this work. The method is supported by statistical analysis. The experimental data support the conclusions. Overall this is an excellent contribution that will interest mycotoxin researchers, food scientists, analytical scientists, chemists, and food safety experts.

 Specific comments to be addressed:

The introduction could be expanded to include a general review aflatoxin B1 detection methods that are approved by the regulatory community.

The merits and limitations of the method could be discussed.

The discussion can benefit by comparing these results to other recent studies on aptasensors for mycotoxins. Specifically, the discussion of these results with respect to the following papers can expand the impact of this manuscript, and make it more relevant to the audience of the journal:

Xu, J.; Liu, J.; Li, W.; Wei, Y.; Sheng, Q.; Shang, Y. A Dual-Signaling Electrochemical Aptasensor Based on an In-Plane Gold Nanoparticles–Black Phosphorus Heterostructure for the Sensitive Detection of Patulin. Foods 2023, 12, 846. https://doi.org/10.3390/foods12040846

 Wang, H.; Chen, L.; Li, M.; She, Y.; Zhu, C.; Yan, M. An Alkyne-Mediated SERS Aptasensor for Anti-Interference Ochratoxin A Detection in Real Samples. Foods 2022, 11, 3407. https://doi.org/10.3390/foods11213407

Guo, W.; Yang, H.; Zhang, Y.; Wu, H.; Lu, X.; Tan, J.; Zhang, W. A Novel Fluorescent Aptasensor Based on Real-Time Fluorescence and Strand Displacement Amplification for the Detection of Ochratoxin A. Foods 2022, 11, 2443. https://doi.org/10.3390/foods11162443

Yao, X.; Yang, Q.; Wang, Y.; Bi, C.; Du, H.; Wu, W. Dual-Enzyme-Based Signal-Amplified Aptasensor for Zearalenone Detection by Using CRISPR-Cas12a and Nt.AlwI. Foods 2022, 11, 487. https://doi.org/10.3390/foods11030487

Author Response

An Impedimetric PAni-based aptasensor was developed to detect aflatoxin B1 in food and feed in various commodities. The selectivity for AFB1 was established. The method had a responsible limit of detection and limit of quantitation. In addition, its cross-reactivities with OTA and AFB2 were determined.  The materials developed have been fully characterized. The introduction covers relevant material but could be expanded. The materials and methods are clearly described and reproducible. The discussion section is comprehensive but could be expanded to discuss work pertinent to this work. The method is supported by statistical analysis. The experimental data support the conclusions. Overall this is an excellent contribution that will interest mycotoxin researchers, food scientists, analytical scientists, chemists, and food safety experts. Specific comments to be addressed:

1) The introduction could be expanded to include a general review aflatoxin B1 detection methods that are approved by the regulatory community.

ANSWER- Mycotoxins are toxic compounds produced as secondary metabolites by fungi, especially by saprophytic molds growing on foodstuffs or animal feeds. Among all the mycotoxins, aflatoxin B1 (AFB1), the carcinogenic product belonging to the flavus, parasiticus and nomius species of the genus Aspergillus is considered the most potent, naturally occurring carcinogen among all the aflatoxins [1,2]. In view of the nature of mycotoxin characteristics, which are polar and nonvolatile, high-performance liquid chromatography (HPLC) is commonly used as the standard analytical method coupled with conventional detectors, such as fluorescence (FLD), UV-visible (UV), photodiode array (PDA), and mass spectrometry (MS) [single MS, and tandem MS (MS/MS)] are employed in HPLC mycotoxin analysis [3]. In view of the limitations associated with official chromatography instrumental analytical techniques, growing demand for small devices and rapid determinations preferably in situ conditions has prompted the development of several probes for AFB1 detection, such as electrochemical immunosensor [4], enzymatic multiwalled carbon nanotubes based on aflatoxin oxidase [5], and a few aptamer-based strategies e.g., fluorescence resonance energy transfer (FRET) biosensor using cadmium telluride (CdTe) quantum dots [1], competitive dipstick assay [6], electrochemical impedimetric assay [7], etc. However, they are marked by poor precision, low sensitivity, stability, and selectivity. The correction has been done in the introduction section, lines 32-49 of the revised manuscript.

New references:

[2] Xiao M.; Bai X.; Liu Y.; Yang L.; Liao X. Simultaneous determination of trace aflatoxin B1 and ochratoxin A by aptamer-based microchip capillary electrophoresis in food samples. J. chromatogr. A 2018, 1569, 222-228. doi: https://doi.org/10.1016/j.chroma.2018.07.051.

[3] Agriopoulou, S.; Stamatelopoulou, E.; Varzakas, T. Advances in analysis and detection of major mycotoxins in foods. Foods 2020, 9, 518. doi: 10.3390/foods9040518.

[4] Abnous, K.; Danesh N.M.; Alibolandi M.; Ramezani M.; Sarreshtehdar Emrani A.; Zolfaghari R.; Taghdisi S.M. A new amplified π-shape electrochemical aptasensor for ultrasensitive detection of aflatoxin B1. Biosens. Bioelectron. 2017, 94, 374–379. doi: 10.1016/j.bios.2017.03.028.

[5] Li S.C.; Chen J.H.; Cao H.; Yao D.S.; Liu D.L. Amperometric biosensor for aflatoxin B1 based on aflatoxin-oxidase immobilized on multiwalled carbon nanotubes. Food Control 2011, 22, 43-49. doi: https://doi.org/10.1016/j.foodcont.2010.05.005.

[6] Shim W.; Kim M.J.; Mun H.; Kim M. An aptemer-based dipstick assay for the rapid and simple detection of aflatoxin B1. Biosens. Bioelectrocn. 2014, 62, 288-294. doi: 10.1016/j.bios.2014.06.059. 

2) The merits and limitations of the method could be discussed. The discussion can benefit by comparing these results to other recent studies on aptasensors for mycotoxins. Specifically, the discussion of these results with respect to the following papers can expand the impact of this manuscript, and make it more relevant to the audience of the journal:

Xu, J.; Liu, J.; Li, W.; Wei, Y.; Sheng, Q.; Shang, Y. A Dual-Signaling Electrochemical Aptasensor Based on an In-Plane Gold Nanoparticles–Black Phosphorus Heterostructure for the Sensitive Detection of Patulin. Foods 2023, 12, 846. https://doi.org/10.3390/foods12040846

 Wang, H.; Chen, L.; Li, M.; She, Y.; Zhu, C.; Yan, M. An Alkyne-Mediated SERS Aptasensor for Anti-Interference Ochratoxin A Detection in Real Samples. Foods 2022, 11, 3407. https://doi.org/10.3390/foods11213407

Guo, W.; Yang, H.; Zhang, Y.; Wu, H.; Lu, X.; Tan, J.; Zhang, W. A Novel Fluorescent Aptasensor Based on Real-Time Fluorescence and Strand Displacement Amplification for the Detection of Ochratoxin A. Foods 2022, 11, 2443. https://doi.org/10.3390/foods11162443

 Yao, X.; Yang, Q.; Wang, Y.; Bi, C.; Du, H.; Wu, W. Dual-Enzyme-Based Signal-Amplified Aptasensor for Zearalenone Detection by Using CRISPR-Cas12a and Nt.AlwI. Foods 2022, 11, 487. https://doi.org/10.3390/foods11030487

ANSWER- Although the detection range of the PAni-based aptasensor (0.03-0.08 nM) is lower than the MLs of AFB1 for foodstuffs and feedstuffs (0.32-64.05 nM), it is a highly sensitive screening technique for the monitoring of AFB1 content in foodstuffs and feedstuffs. Therefore, it is ideal for the early and sensitive detection of AFB1 contamination and appropriate precautions can be taken to avoid reaching the MLs of AFB1 in the foodstuffs and feedstuffs. For real samples with AFB1 concentration above the upper limit of quantitation of the proposed PAni-based impedimetric aptasensor, appropriate dilution of the sample using 0.01 M PBS as the diluent is needed before detection to minimize the background noise.

To improve the sensitivity over a wide range of analyte mycotoxin, recent studies on aptasensor with advanced and emerging techniques have been reported based on an in-plane AnNPs–black phosphorus heterostructure (AuNPs-BPNS) was synthesized for signal amplification with dual-signaling strategy, in which a methylene-blue-labeled aptamer and ferrocene monocarboxylic acid in the electrolyte was fabricated for sensitive detection of patulin [71]. Yao, et al. [72] developed a dual-enzyme-based signal-amplified aptasensor with emerging CRISPR-based technology to sensitively detect zearalenone by using two aptamers as recognition elements and EnGen LbaCas12a and Nt.AlwI nicking endonuclease as signal amplifiers. Guo’s research group [73], on the other hand, has made use of strand displacement amplification primers as an amplification approach initiated under the action of Bst DNA polymerase and nicking endonuclease for real-time fluorescence aptasensing of ochratoxin A with amplified ssDNA products dyed with SYBR Green II. A highly sensitive and specific aptasensor has been developed by Wang et al., [74] for the anti-interference detection of OTA. 4-[(Trimethylsilyl) ethynyl] aniline (4-TEAE) and OTA-aptamer were assembled on the AuNPs to serve as anti-interference SERS probes with Fe3O4 nanoparticles linked to the complementary aptamer. The specific binding of OTA to aptamer inhibited the binding of SERS probes and capture probes and attenuated the Raman responses. Corrections have been done to lines 607-610 and lines 732-749 of the revised manuscript.

New references:

[71] Xu, J.; Liu, J.; Li, W.; Wei, Y.; Sheng, Q.; Shang, Y. A dual-signaling electrochemical aptasensor based on an in-plane gold nanoparticles–black phosphorus heterostructure for the sensitive detection of patulin. Foods 2023, 12, 846. doi: https://doi.org/10.3390/foods12040846.

[72] Yao, X.; Yang, Q.; Wang, Y.; Bi, C.; Du, H.; Wu, W. Dual-enzyme-based signal-amplified aptasensor for zearalenone detection by using CRISPR-Cas12a and Nt.AlwI. Foods 2022, 11, 487. doi: https://doi.org/10.3390/foods11030487.

[73] Guo, W.; Yang, H.; Zhang, Y.; Wu, H.; Lu, X.; Tan, J.; Zhang, W. A Novel Fluorescent Aptasensor Based on Real-Time Fluorescence and Strand Displacement Amplification for the Detection of Ochratoxin A. Foods 2022, 11, 2443. doi: https://doi.org/10.3390/foods11162443.

[74] Wang, H.; Chen, L.; Li, M.; She, Y.; Zhu, C.; Yan, M. An alkyne-mediated SERS aptasensor for anti-interference ochratoxin A detection in real samples. Foods 2022, 11, 3407. doi: https://doi.org/10.3390/foods11213407.
